# Lifestyle parameters of Japanese agricultural and non-agricultural workers aged 60 years or older and less than 60 years: A cross-sectional observational study

**Fumie Kinoshita[1], Kei Yoshida[1,2], Masaya Fujitani[1,2], Yuta Imai[1,2], Yumiko Kobayashi[1], Tomoya Ito[2,3], Yuto Okumura[2,3], Hiroyuki Sato[4], Tatsuya Mikami[5], Songee Jung[5,6], Akihiro Hirakawa[4]\*, Masahiro Nakatochi[3]\***

1 Department of Advanced Medicine, Data Science Division, Data Coordinating Center, Nagoya University Hospital, Nagoya, Aichi, Japan, 2 Department of Basic Medicinal Sciences, Graduate School of Pharmaceutical Sciences, Nagoya University, Nagoya, Aichi, Japan, 3 Department of Integrated Health Sciences, Public Health Informatics Unit, Nagoya University Graduate School of Medicine, Nagoya, Aichi, Japan, 4 Department of Clinical Biostatistics, Graduate School of Medical and Dental Sciences, Tokyo Medical and Dental University, Tokyo, Japan, 5 Innovation Center for Health Promotion, Hirosaki University Graduate School of Medicine, Hirosaki, Aomori, Japan, 6 Department of Environmental Health Science and Public Health, Akita University Graduate School of Medicine, Akita, Akita, Japan

\* mnakatochi@met.nagoya-u.ac.jp (MN); a-hirakawa.crc@tmd.ac.jp (AH)

## Abstract

### Objectives

Improving the lifestyle of occupational workers is essential for extending healthy life expectancy. We investigated various lifestyle-related items in a rural Japanese population and compared them between agricultural and non-agricultural workers.

### Methods

This cross-sectional study was conducted as a part of the "Iwaki Health Promotion Project." Lifestyle-related items such as sleep, work hours, nutrition, health-related quality of life, and proportion of time spent performing each daily activity were compared between agricultural and non-agricultural workers in the ≥60 years (n = 251) and <60 years (n = 560) age groups.

### Results

Agricultural workers had significantly lower Pittsburgh Sleep Quality Index total scores than non-agricultural workers in the <60 years group. The proportion of participants with more than 5 weekly working days was high among agricultural workers in both groups. Additionally, the proportion of people who worked more than 8 h per day was high among agricultural workers in both age groups. Energy intake per day was high among agricultural workers in the <60 years group. In both age groups, agricultural workers slept and woke up approximately 40 min earlier than did non-agricultural workers.

**Data Availability Statement:** The datasets generated during and/or analyzed during the current study are not publicly available because of ethical concerns but are available from the Hirosaki

University COI Program Institutional Data Access/ Ethics Committee (contact via e-mail: coi@hirosaki-u.ac.jp) for researchers who meet the criteria for accessing confidential data. Researchers need to be approved by a research ethics review at the organization with which the researchers using the data are affiliated.

**Funding:** MN and AH: The Center of Innovation Program from the Japan Science and Technology Agency (JPMJCE1302, JPMJCA2201). https:// www.jst.go.jp/tt/EN/platform/coi.html FK: JSPS KAKENHI (20K18848). https://www.jsps.go.jp/ english/ The funders had no role in study design, data collection and analysis, decision to publish, or preparation of the manuscript.

**Competing interests:** The authors have declared that no competing interests exist.

## Conclusions

Agricultural workers have better sleep habits but work longer than non-agricultural workers, with some differences in energy intake and proportion of time spent on each daily activity. These differences should be considered when planning lifestyle intervention programs for agricultural workers.

## Introduction

In Japan, the proportion of people aged over 65 years reached 28.9% in 2021 [1]. Extending healthy life expectancies has become a pressing issue. To achieve this, a healthy lifestyle is considered effective [2–7]. Japan's Ministry of Health, Labour and Welfare formulated the Health Japan 21 (the second term) plan [8] and conducted various projects [9] to comprehensively promote people's health. One target is to improve everyday habits and social environment factors related to nutrition and dietary habits, physical activity and exercise, rest, alcohol consumption, smoking, and dental and oral health. In this plan, utilizing various channels, such as mass media including ICT, volunteer groups relating to health promotion, and industry, and providing practical encouragement that meets the characteristics of the target group are essential [8].

Since occupation is considered to significantly impact lifestyle differences, clarifying these differences by occupation may be useful for implementing optimal and effective interventions [10,11]. Agricultural workers may have lifestyles different from people in other occupations [12–14] because their work style differs greatly from that of those who work in companies. Furthermore, many Japanese agricultural workers own family-run businesses and are less likely to undergo health checkups and interventions compared with company workers [15,16].

Health outcomes differ among agricultural workers. Compared with non-agricultural workers, agricultural workers have a longer life expectancy [14,17], less long-term care prior to death [18], lower overall risk of cancer [19], and higher risk for mental health problems [20,21]. These differences are associated with lifestyle differences [12,14,22]. However, limited reports have examined the diverse range of lifestyle factors among agricultural and non-agricultural workers in Japan. A study conducted by Ohta explored the varying lifestyles within occupational groups, including agricultural and forestry workers. However, it is worth noting that the study primarily focused on evaluating men in their 40s and 50s. Further investigation is needed to gain a comprehensive understanding of the lifestyle factors among workers across different age groups and genders in both agricultural and non-agricultural sectors in Japan [11]. Zheng reported a difference in health-promoting lifestyles but only evaluated these using the Health Promoting Lifestyle Profile II (HPLC-II) [10].

It would be meaningful to assess agricultural worker's specific lifestyles across a wide range of working ages and genders, as this would provide basic data for considering how the government and companies can intervene effectively and designing future prospective studies to assess the effect of interventions. In this hypothesis-generating exploratory study, we investigated various lifestyle-related items in a rural Japanese population and compared them between agricultural and non-agricultural workers to guide future interventions for occupational workers.

## Materials and methods

### Participants and data collection

This hypothesis-generating cross-sectional study is part of an ongoing observational study called the "Iwaki Health Promotion Project" that began in 2005, which entails an annual large-

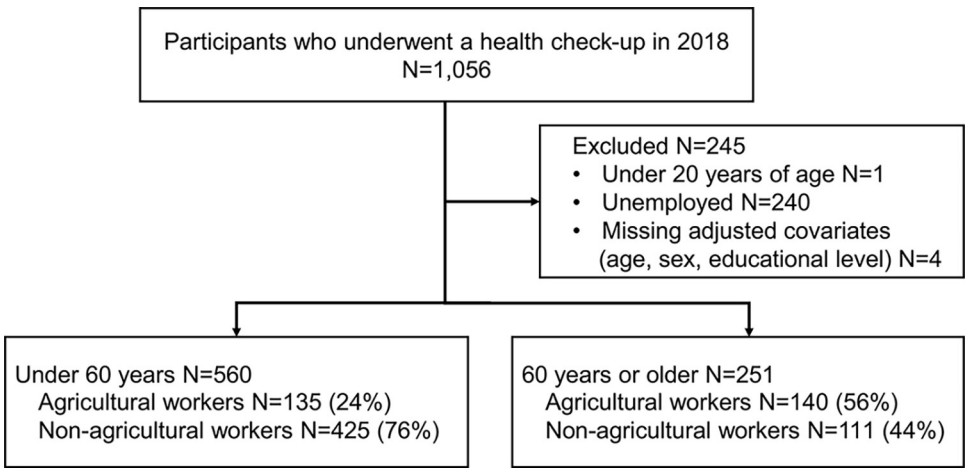

**Fig 1. Flow diagram of participants.**

scale health survey of the residents of Iwaki-ku (formerly Iwaki-cho) in Hirosaki City, Aomori Prefecture, Japan [23]. Data from 2018 were used because it was the first year in which agricultural work was included as a main occupation. Among residents of the Iwaki district of Hirosaki City (comprising approximately 6000 people), 1,056 voluntary participants were included in the study (Fig 1) [23]. Participants aged <20 years, or those with missing data for at least one of the following factors, were excluded from the analysis: age, sex, or education.

Participants were divided into age groups of <60 and ≥ 60 to account for occupational changes that may be related to Japan's typical retirement age of 60 years. Occupation data were collected using a questionnaire that began with "What has been your main occupation (including homemaker) during the past year?" Overall, 14 options were obtained, including the 11 shown in Table A in S1 Appendix, plus "homemaker," "unemployed," and "student." Participants who answered that their main occupation during the past year was agricultural work were defined as agricultural workers, whereas those who responded that their main occupation was other were classified as non-agricultural workers. The 240 people who selected "homemaker," "unemployed," or "student" were excluded.

This study was approved by Hirosaki University School of Medicine (2021–166), Nagoya University School of Medicine (2016–0137), and Tokyo Medical and Dental University (M2020-186). All participants provided written informed consent.

## Measurements

Body mass index (BMI) and body fat percentage were measured during the health checkup. Data on working, sleeping, and other lifestyle factors were collected through a questionnaire. For the proportion of time spent on each daily activity, the number of days and hours per week that the participant spent performing each activity were collected through a questionnaire, and the percentage of time spent on each daily activity per week was calculated. Data were collected in 18 categories as described in the S2 Appendix. Health-Related Quality of Life (HRQoL) was assessed using the Short Form Health Survey version 2 (SF-36 v2). The SF-36 is a widely used, self-administered health questionnaire comprising 36 items on eight subscales [24]. We used the Japanese version of SF-36. Each scale was represented by a score, with higher scores indicating higher HRQoL [25,26]. To assess sleep quality, participants were assessed using the Japanese version of the Pittsburgh Sleep Quality Index (PSQI) developed by Buysse

et al. [27,28] The higher the score, the worse the sleep quality. To assess nutritional status, participants were surveyed using the brief self-administered diet history questionnaire (BDHQ) [29,30]. This questionnaire is described in S2 Appendix.

## Statistical analysis

The median, first, and third quartile for continuous variables and frequencies (%) for categorical variables were used. Characteristics were compared between agricultural and non-agricultural workers using the Wilcoxon rank-sum test for continuous variables and Fisher's exact test for categorical variables.

A linear regression model for continuous variables and a logistic regression model for categorical variables were applied to compare the lifestyles of agricultural and non-agricultural workers. Firth bias-corrected logistic regression was used if the logistic regression failed to converge. Lifestyle-related variables were the dependent variables, and occupation (1: agricultural workers, 0: non-agricultural workers) was the independent variable. Results were adjusted for age as a categorical variable because the dependent variables may not be linearly related to age (20–39, 40–49, 50–59 in the <60 years group, and 60–70, ≥70 in the ≥60 years group), sex, and years of education (9–11, 12–13, 14–15, ≥16, and other). No multiple testing correction was performed because all hypotheses were considered independently [31].

The mean and distribution of bedtime and wake-up time were calculated by R package "circular" version 0.4–93. For comparisons, we used the circular general linear model by R package "bpnreg" version 2.0.2 with the Markov Chain Monte Carlo algorithm [32,33], adjusted for age, sex, and years of education. Variables were considered to have an effect if at least one of the 95% highest probability density (HPD) intervals for component I and component II linear coefficients did not include 0.

A complete case analysis was used because missing data were within 5% of the total [34].

All statistical tests were two-sided. Statistical significance was set at P <0.05. All analyses were performed using SAS version 9.4 (SAS Institute, Cary, NC, USA) and R version 4.1.2.

## Results

### Participant characteristics

Among 1,056 participants in the 2018 survey, 251 aged ≥60 and 560 aged <60 (Fig 1) were included. Participants were surveyed about their main occupation, and the proportion of each is shown in Table A in S1 Appendix. The proportion of agricultural workers was 24.1% in the <60 years group and 55.8% in the ≥60 years group. The participants' characteristics are shown in Table 1. In the <60 years group, the median (interquartile range [IQR]) age was 45.0 (37.0–56.0) years for agricultural workers and 42.0 (35.0–50.0) years for non-agricultural workers, being significantly higher for agricultural workers (p<0.001) (Fig 2). In the ≥60 years group, the median (IQR) age was 68.0 years (64.0–72.0) for agricultural workers and 65.0 years (62.0–68.0) for non-agricultural workers, being significantly higher for agricultural workers (p<0.001) (Fig 2). In the <60 years group, the number (proportion) of males was 78 (57.8%) among agricultural workers and 185 (43.5%) among non-agricultural workers, being significantly higher for agricultural workers (p = 0.004). In the <60 years age group, the median (IQR) BMI was 22.70 (20.90–25.20) among agricultural workers and 22.10 (19.90–24.50) among non-agricultural workers, being significantly higher in agricultural workers (p = 0.017). In the ≥60 years group, the education level was higher among non-agricultural workers than among agricultural workers (p<0.001).

**Table 1. Participant characteristics according to agricultural and non-agricultural workers.**

| | | Age <60 years | | | Age ≥60 years | | |
|---|---|---|---|---|---|---|---|
| | | Agricultural workers n = 135 | Non-agricultural workers n = 425 | | Agricultural workers n = 140 | Non-agricultural workers n = 111 | |
| | | Median (IQR) or n (%) | | p-value* | Median (IQR) or n (%) | | p-value* |
| Age (years) | | 45.0 (37.0–56.0) | 42.0 (35.0–50.0) | **<0.001** | 68.0 (64.0–72.0) | 65.0 (62.0–68.0) | **<0.001** |
| Sex | Male | 78 (57.8) | 185(43.5) | **0.004** | 68(48.6) | 57(51.4) | 0.704 |
| BMI (kg/m$^2$) | | 22.70 (20.90–25.20) | 22.10 (19.90–24.50) | **0.017** | 23.30 (21.40–25.60) | 23.00 (20.80–25.50) | 0.450 |
| Body fat percentage (%) | | 22.50 (17.40–28.20) | 24.20 (19.40–29.75) | 0.069 | 25.60 (20.80–31.50) | 25.00 (20.60–31.40) | 0.9494 |
| Education (years) | 6–11 | 5 (3.7) | 7(1.6) | 0.068 | 43(30.7) | 20(18.0) | **<0.001** |
| | 12–13 | 85 (63.0) | 238(56.0) | | 76(54.3) | 51(45.9) | |
| | 14–15 | 35 (25.9) | 115(27.1) | | 15(10.7) | 27(24.3) | |
| | ≥16 | 10 (7.4) | 64(15.1) | | 4(2.9) | 13(11.7) | |
| | Other | 0 (0) | 1(0.2) | | 2(1.4) | 0(0) | |

\* P-value was determined using Wilcoxon rank-sum test or Fisher's exact test. Boldface indicates statistical significance.

IQR, interquartile range.

## Differences in lifestyle between agricultural and non-agricultural workers

Lifestyle comparisons among workers are presented in Tables 2 and 3. The proportion of participants with a sleeping disorder was lower among agricultural workers than among non-agricultural workers in the <60 years group (odds ratio [OR] [vs non-agricultural workers] = 0.49, 95% CI = 0.27–0.88, p = 0.016); however, there was no significant difference in the ≥60 years group (OR = 1.32, 95% CI = 0.59–2.98, p = 0.502). The proportion of participants who took a nap was high among agricultural workers in both age groups (OR = 3.67, 95% CI = 2.37–5.67, p<0.001 in the <60 years group; OR = 2.28, 95% CI = 1.30–3.99, p = 0.004 in the ≥60 years group). Energy intake per day was high among agricultural workers in the <60 years group (beta [reference was non-agricultural workers] = 158 kcal/day, 95% CI = 54–262, p = 0.003).

The proportion of participants who worked more than 5 days a week was high among agricultural workers in both age groups (OR = 17.14, 95% CI = 8.42–34.89, p<0.001 in the <60 years group; OR = 3.37, 95% CI = 1.86–6.10, p<0.001 in the ≥60 years group); similarly, the proportion of people whose working time per day was more than 8 h was high among agricultural workers in both age groups (OR = 2.14, 95% CI = 1.38–3.31, p = 0.001 in the <60 years group; OR = 2.40, 95% CI = 1.31–4.40, p = 0.005 in the ≥60 years group). There were no clear differences in exercise, current smoking, and current drinking habits between agricultural and non-agricultural workers.

The results of SF-36 showed that the bodily pain (BP) score was significantly lower for agricultural workers <60 years of age (beta = -2.1, 95% CI = -3.9–-0.2, p = 0.029). The score for the role limitations attributable to emotional problems was significantly lower for agricultural workers ≥60 years of age (beta = -2.4, 95% CI = -4.6–-0.1, p = 0.037).

The proportion of time spent on each daily activity showed that personal chores (beta = -0.7, 95% CI = -1.1–-0.3, p = 0.002), eating (beta = -0.8, 95% CI = -1.3–-0.3, p = 0.003), commuting to work or school (beta = -1.1, 95% CI = -1.6–-0.5, p<0.001), traveling (beta = -0.8, 95% CI = -1.4–-0.3, p = 0.004), and active leisure activities (beta = -0.9, 95% CI = -1.8–-0.1, p = 0.036) were significantly lower, and passive leisure activities (beta = 3.0, 95% CI = 0.7–5.3, p = 0.001) were higher in agricultural workers, respectively, in the <60 years group. Time

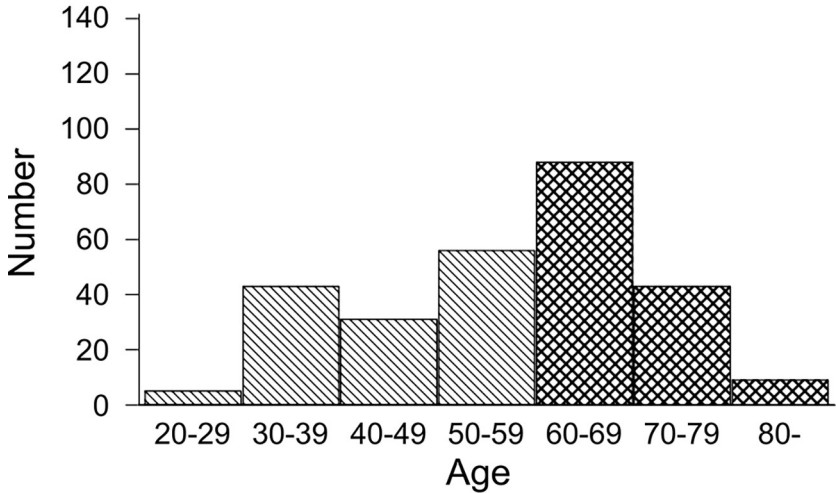

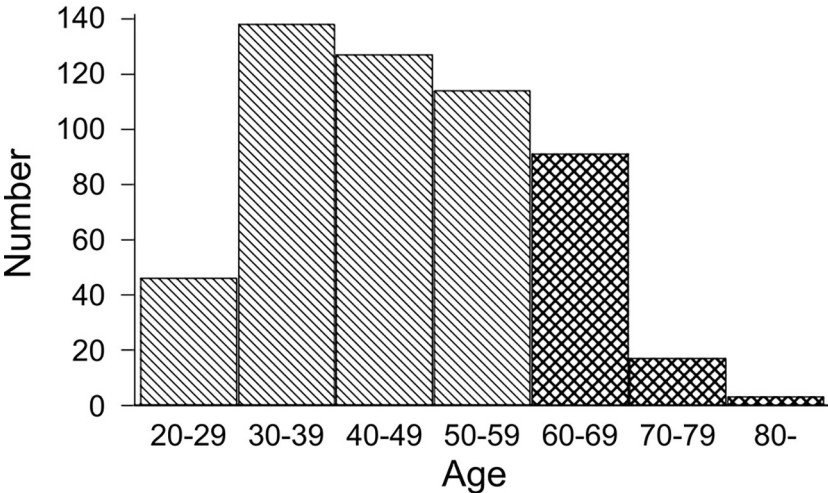

**Fig 2. Distribution of age in agricultural and non-agricultural workers.** Diagonal line areas indicate age <60 years, whereas shaded areas indicate age ≥60 years.

spent socializing (beta = -0.7, 95% CI = -1.1–-0.2, p = 0.002) was significantly lower in the ≥60 years group. In both age groups, the participation rate in hobbies and amusements (OR = 0.58, 95% CI = 0.38–0.90, p = 0.014 in the <60 years group; OR = 0.45, 95% CI = 0.26–0.80, p = 0.006 in the ≥60 years group) and sports (OR = 0.51, 95% CI = 0.28–0.94, p = 0.030 in the <60 years group; OR = 0.42, 95% CI = 0.20–0.88, p = 0.021 in the ≥60 years group) was significantly lower among agricultural workers.

## Difference in bedtime and wake-up time among occupations

The proportion of participants' bedtime and wake-up time for each hour of the day is shown in Fig 3A and 3B, respectively, and were compared with the mean time for agricultural and non-agricultural workers (Table 4). In both age groups, agricultural workers' mean bedtime

**Table 2. Comparison of continuous lifestyle variables between agricultural and non-agricultural workers.**

| | Age <60 years | | | | Age ≥60 years | | | |
|---|---|---|---|---|---|---|---|---|
| | Agricultural workers | Non-agricultural workers | | | Agricultural workers | Non-agricultural workers | | |
| | Mean (95% CI) | Mean (95% CI) | Adjusted β (95% CI)* | p-value** | Mean (95% CI) | Mean (95% CI) | Adjusted β (95% CI)* | p-value** |
| **Sleep** | | | | | | | | |
| PSQI total score | 3.3 (2.9–3.6) | 4.0 (3.8–4.2) | -0.6 (-1.1--0.2) | **0.005** | 3.5 (3.1–3.9) | 3.5 (3.0–3.9) | 0.1 (-0.5–0.8) | 0.652 |
| Sleeping hours (hours) | 6.8 (6.5–7.1) | 6.8 (6.6–7.0) | -0.1 (-0.5–0.3) | 0.728 | 7.1 (6.8–7.5) | 7.0 (6.7–7.3) | 0.0 (-0.5–0.4) | 0.997 |
| **BDHQ** | | | | | | | | |
| Energy intake (kcal/day) | 2045 (1941–2149) | 1811 (1758–1864) | 158 (54–262) | **0.003** | 1993 (1899–2087) | 1942 (1839–2045) | 24 (-106–155) | 0.714 |
| Percentage of protein (%) | 14.3 (13.9–14.7) | 14.6 (14.3–14.9) | -0.1 (-0.6–0.5) | 0.817 | 15.3 (14.8–15.8) | 16.3 (15.8–16.9) | -1.0 (-1.7--0.2) | **0.012** |
| Percentage of lipid (%) | 24.6 (23.6–25.6) | 26.7 (26.1–27.3) | -1.3 (-2.4--0.1) | **0.028** | 23.9 (23.0–24.9) | 26.9 (25.7–28.0) | -2.4 (-3.9--1.0) | **0.001** |
| Percentage of carbohydrate (%) | 53.5 (52.1–54.9) | 51.7 (50.9–52.5) | 2.0 (0.4–3.6) | **0.017** | 54.8 (53.5–56.1) | 51.5 (49.7–53.3) | 2.6 (0.4–4.9) | **0.023** |
| Salt intake (g/day) | 11.2 (10.6–11.9) | 10.2 (9.9–10.5) | 0.6 (0.0–1.3) | 0.058 | 12.0 (11.4–12.5) | 12.2 (11.5–13.0) | -0.6 (-1.4–0.3) | 0.225 |
| **SF-36** | | | | | | | | |
| Physical functioning | 54.1 (52.5–55.7) | 54.8 (54.0–55.5) | -0.2 (-1.8–1.4) | 0.834 | 45.8 (43.6–48.0) | 47.6 (45.0–50.2) | -0.3 (-3.7–3.1) | 0.865 |
| Role physical | 53.3 (52.4–54.3) | 53.8 (53.2–54.4) | -0.3 (-1.5–1.0) | 0.690 | 48.8 (46.9–50.8) | 50.9 (49.3–52.5) | -1.7 (-4.2–0.9) | 0.200 |
| Bodily pain | 48.9 (47.3–50.5) | 51.2 (50.3–52.1) | -2.1 (-3.9--0.2) | **0.029** | 47.3 (45.8–48.9) | 48.9 (47.1–50.7) | -1.3 (-3.8–1.2) | 0.292 |
| General health | 48.8 (47.3–50.3) | 49.6 (48.7–50.5) | -0.3 (-2.1–1.5) | 0.762 | 46.8 (45.4–48.1) | 48.1 (46.3–49.8) | -1.3 (-3.6–1.0) | 0.262 |
| Vitality | 50.5 (49.1–51.8) | 48.8 (47.8–49.7) | 1.3 (-0.5–3.1) | 0.163 | 51.0 (49.4–52.6) | 52.8 (51.1–54.5) | -1.9 (-4.3–0.6) | 0.135 |
| Social functioning | 52.5 (51.1–54.0) | 53.2 (52.5–53.9) | -0.6 (-2.1–0.9) | 0.464 | 53.4 (52.2–54.6) | 53.9 (52.8–55.1) | -0.8 (-2.5–1.0) | 0.390 |
| Role emotional | 54.4(53.4–55.3) | 53.3(52.5–54.0) | 1.0(-0.4–2.4) | 0.166 | 50.8(49.0–52.6) | 53.7(52.7–54.8) | -2.4(-4.6--0.1) | **0.037** |
| Mental health | 51.9(50.5–53.2) | 49.9(49.0–50.8) | 1.6(-0.2–3.4) | 0.081 | 52.6(51.0–54.3) | 52.6(51.2–54.1) | -0.1(-2.5–2.2) | 0.913 |
| Physical component summary | 54.4(53.1–55.6) | 55.8(55.1–56.4) | -0.9(-2.3–0.5) | 0.189 | 47.5(45.4–49.5) | 49.6(47.6–51.6) | -1.1(-4.0–1.7) | 0.433 |
| Mental component summary | 49.3(47.9–50.6) | 48.0(47.1–48.9) | 0.9(-0.8–2.7) | 0.304 | 51.9(50.4–53.4) | 52.3(50.7–53.9) | -0.8(-3.1–1.4) | 0.469 |
| **Proportion of time spent on each daily activity** | | | | | | | | |
| Personal chores (%) | 2.8(2.5–3.1) | 3.7(3.4–3.9) | -0.7(-1.1--0.3) | **0.002** | 3.0(2.4–3.5) | 3.4(2.7–4.2) | -0.4(-1.3–0.5) | 0.399 |
| Eating (%) | 4.6(4.2–4.9) | 5.5(5.2–5.7) | -0.8(-1.3--0.3) | **0.003** | 4.9(4.4–5.4) | 4.9(4.4–5.4) | 0.0(-0.7–0.7) | 0.959 |
| Commuting to work or school (%) | 1.4(1.1–1.7) | 2.4(2.1–2.7) | -1.1(-1.6--0.5) | **<0.001** | 1.1(0.3–1.8) | 1.3(1.0–1.6) | -0.2(-1.0–0.7) | 0.677 |
| Travel time (excluding commuting to work or school) (%) | 1.4(1.0–1.8) | 2.3(2.0–2.6) | -0.8(-1.4--0.3) | **0.004** | 1.2(0.7–1.7) | 1.6(1.1–2.2) | -0.5(-1.3–0.3) | 0.206 |
| Housework and related work (%) | 10.2(8.1–12.3) | 12.1(10.7–13.4) | 0.4(-1.9–2.7) | 0.722 | 6.7(5.3–8.2) | 7.6(6.1–9.1) | -1.1(-2.8–0.7) | 0.231 |

(*Continued*)

**Table 2.** (Continued)

| | Age <60 years | | | | Age ≥60 years | | | |
|---|---|---|---|---|---|---|---|---|
| | Agricultural workers | Non-agricultural workers | | | Agricultural workers | Non-agricultural workers | | |
| | Mean (95% CI) | Mean (95% CI) | Adjusted β (95% CI)* | p-value** | Mean (95% CI) | Mean (95% CI) | Adjusted β (95% CI)* | p-value** |
| Passive leisure activities (%) | 16.5(14.1–18.8) | 13.1(12.1–14.1) | 3.0(0.7–5.3) | **0.001** | 18.1(15.3–20.9) | 17.8(15.5–20.0) | 0.0(-3.8–3.8) | 0.987 |
| Active leisure activities (%) | 1.8(1.3–2.4) | 2.8(2.3–3.2) | -0.9(-1.8–-0.1) | **0.036** | 2.1(0.9–3.3) | 3.0(1.3–4.7) | -0.7(-2.8–1.4) | 0.504 |
| Socializing (%) | 0.7(0.4–1.0) | 0.9(0.8–1.1) | -0.2(-0.6–0.2) | 0.339 | 0.5(0.3–0.8) | 1.1(0.7–1.4) | -0.7(-1.1–-0.2) | **0.002** |
| Medical examination and treatment (%) | 0.2(0.1–0.3) | 0.2(0.1–0.2) | 0.0(-0.1–0.2) | 0.445 | 0.5(0.2–0.7) | 0.2(0.1–0.3) | 0.2(0.0–0.5) | 0.095 |

Beta value and p-value adjusted for age, sex, and years of education.

* Beta value is a partial regression coefficient with non-agricultural workers as the reference.

** Boldface indicates statistical significance.

CI, confidence interval; PSQI, Pittsburgh Sleep Quality Index; BDHQ, brief self-administered diet history questionnaire.

was approximately 40 min earlier than that of non-agricultural workers, and the HPD interval for at least one component did not include 0. Similarly, agricultural workers woke up earlier.

## Discussion

In this study, we compared the lifestyles of agricultural workers with those of other occupational workers to guide efficient health-promoting lifestyle interventions. We captured the lifestyle characteristics of agricultural workers, who may have a different lifestyle from those of other occupations.

The number of working days per week and working hours per day were longer for agricultural workers in both age groups. This result is in line with national statistics [35,36]. Some studies reported that long working hours negatively affect health outcomes, including an increased association with depression, coronary heart disease, or other mental or physical morbidities [22,37–39]. However, Japanese agriculture often involves family businesses; thus, it is not easy to shorten working hours [16]. Hence, it is better to use one's time outside of work efficiently, since it is limited. In particular, it is important to find ways to quickly and efficiently reduce fatigue.

An assessment of HRQoL data from the SF-36 showed that agricultural workers had greater BP than non-agricultural workers aged <60 years. As agricultural work is physically demanding, rest may be more desirable for reducing pain originating from work than increasing exercise or sports habits during leisure time. In fact, the participation rate in sports and hobbies was significantly lower among agricultural workers in both age groups; furthermore, in the <60 years group, the proportion of time spent on passive leisure activities was greater. Previous reports have stated that sleep during rest periods between work is important for recovery from work-related fatigue, and a recovery training program was also being considered in Germany [40]. Providing such programs and making effective use of leisure time is expected to improve worker safety, recovery from fatigue, and health maintenance for agricultural workers.

Regarding sleep, appropriate sleeping hours and no sleep disturbances are associated with longer healthy and chronic disease-free life expectancy [41]. In our study, the analysis of PSQI

**Table 3. Comparison of categorical lifestyle variables between agricultural and non-agricultural workers.**

| | Age <60 years | | | | Age ≥60 years | | | |
|---|---|---|---|---|---|---|---|---|
| | Agricultural workers | Non-agricultural workers | | | Agricultural workers | Non-agricultural workers | | |
| | n (%) | n (%) | Adjusted OR (95% CI) | p-value* | n (%) | n (%) | Adjusted OR (95% CI) | p-value* |
| **Sleep** | | | | | | | | |
| Sleeping disorder | 16(12.1) | 94(22.3) | 0.49(0.27–0.88) | **0.016** | 17(13.8) | 13(12.3) | 1.32(0.59–2.98) | 0.502 |
| Napping | 76(58.0) | 108(25.5) | 3.67(2.37–5.67) | **< .001** | 77(60.2) | 42(38.2) | 2.28(1.30–3.99) | **0.004** |
| **Working** | | | | | | | | |
| Weekly working days >5 days | 125(93.3) | 177(42.1) | 17.14(8.42–34.89) | **< .001** | 109(80.1) | 59(53.6) | 3.37(1.86–6.10) | **< .001** |
| Working time per day >8 h | 65(48.5) | 122(29.2) | 2.14(1.38–3.31) | **0.001** | 57(42.5) | 29(26.9) | 2.40(1.31–4.40) | **0.005** |
| **Habit** | | | | | | | | |
| Regular physical activity or exercise | 25(18.5) | 94(22.1) | 0.74(0.45–1.24) | 0.253 | 30(21.6) | 33(29.7) | 0.75(0.40–1.39) | 0.363 |
| Current smoking | 36(26.9) | 97(22.9) | 1.00(0.62–1.60) | 0.985 | 9(6.5) | 11(10.1) | 0.67(0.26–1.74) | 0.414 |
| Current drinking | 83(61.5) | 214(51.2) | 1.18(0.77–1.80) | 0.46 | 64(47.4) | 48(44.9) | 1.19(0.63–2.24) | 0.585 |
| **Social environment** | | | | | | | | |
| Presence of a spouse | 101(75.9) | 296(70.0) | 1.15(0.72–1.84) | 0.558 | 100(78.1) | 87(79.1) | 1.48(0.71–3.06) | 0.293 |
| **Daily activity** | | | | | | | | |
| Learning/self-education/training | 17(12.9) | 61(14.6) | 0.91(0.50–1.65) | 0.745 | 13(10.6) | 20(18.2) | 0.79(0.35–1.79) | 0.567 |
| Hobbies and amusements | 53(40.5) | 216(51.3) | 0.58(0.38–0.90) | **0.014** | 37(30.1) | 53(48.6) | 0.45(0.26–0.80) | **0.006** |
| Sports | 15(11.5) | 79(18.9) | 0.51(0.28–0.94) | **0.03** | 16(12.7) | 28(25.7) | 0.42(0.20–0.88) | **0.021** |
| **Dental and oral health** | | | | | | | | |
| Regular dental checkups | 75(56.4) | 244(57.8) | 1.01(0.67–1.51) | 0.974 | 77(60.2) | 69(62.7) | 0.96(0.55–1.67) | 0.884 |
| Brush teeth ≥2 times per day | 104(78.2) | 364(86.5) | 0.76(0.44–1.30) | 0.315 | 97(75.8) | 84(76.4) | 0.88(0.46–1.72) | 0.715 |

Odds ratio and p-value adjusted for age, sex, and years of education.

* Boldface indicates statistical significance.

OR, odds ratio; CI, confidence interval.

and sleeping disorder prevalence revealed that agricultural workers were getting better quality sleep than non-agricultural workers in the <60 years group. A similar result was previously reported in China, wherein farmers had longer sleep duration and better sleep quality according to the PSQI than did other occupational workers [42]. Conversely, there was no significant difference in PSQI score and sleeping disorder prevalence among those aged ≥60 years. Regarding sleep duration, no significant differences were found in either age group in this study, although shorter sleep duration among blue-collar workers has been reported [43]. This difference may be due to the fact that their definition of blue-collar workers included non-agricultural workers such as security or transport workers. In both age groups, agricultural

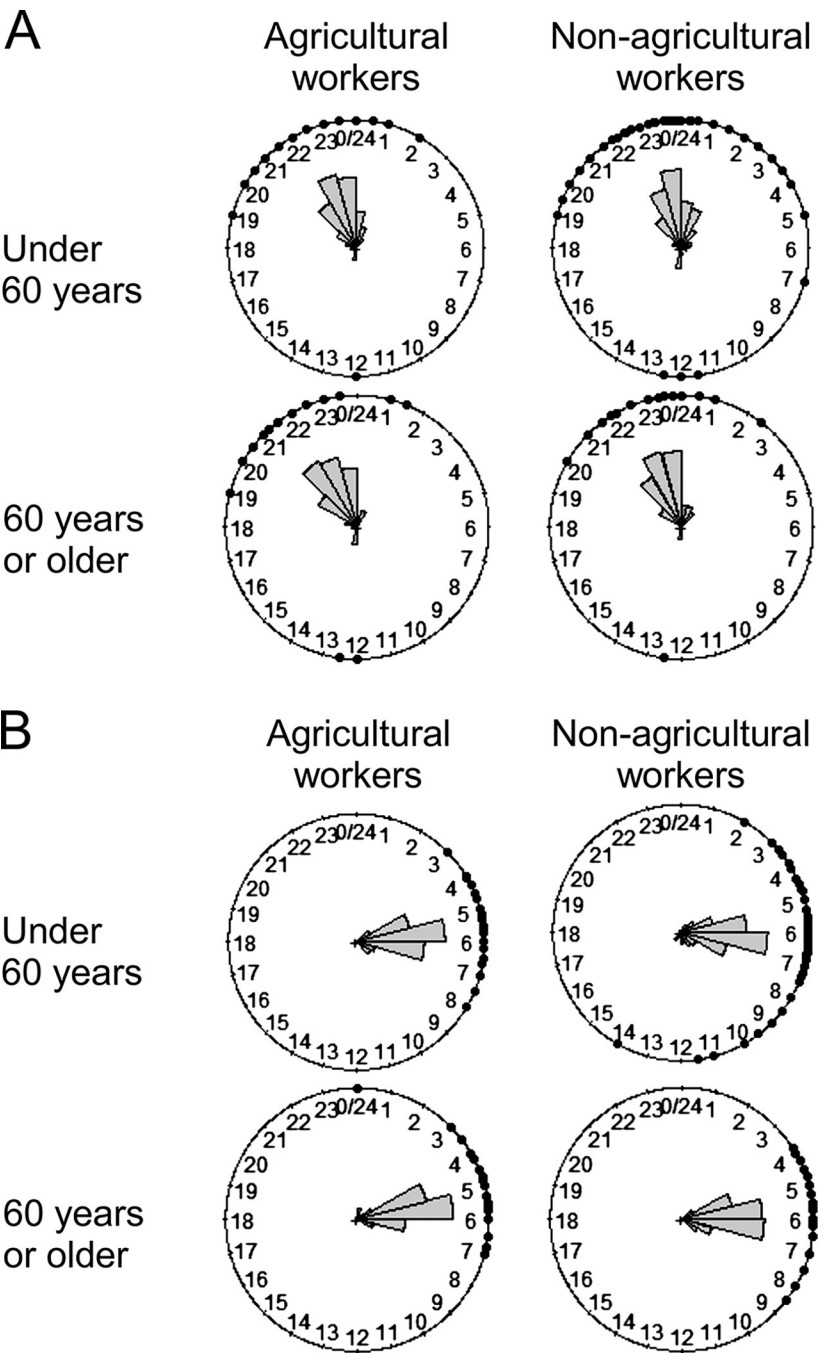

**Fig 3. Proportion of participants who went to bed and woke up at each hour of the day.** (a) A circular histogram shows bedtime and (b) wake-up times. The gray bar shows the proportion of people per hour. The plots show the data points.

workers tended to go to bed earlier and wake up earlier than non-agricultural workers. Some reports show that going to bed early and rising early is good for health; for example, people who stayed up late and got up late were more likely to visit doctors [44]. Maintaining good sleep among young agricultural workers is desirable for a healthy life, and it would be helpful to maintain this habit.

**Table 4. Comparison of mean bedtime and wake-up time between agricultural and non-agricultural workers.**

|  | Age group | Agricultural workers | Non-agricultural workers |
|---|---|---|---|
| **Bedtime** | Age <60 years* | 22:28 | 23:16 |
|  | Age ≥60 years* | 21:45 | 22:26 |
| **Wake-up time** | Age <60 years* | 5:21 | 6:04 |
|  | Age ≥60 years* | 5:01 | 5:36 |

The circular general linear model was performed by adjusting for age, sex, and years of education.

*Asterisks indicate that the highest probability density interval of difference was not included.

The results of this study showed that energy intake was higher among agricultural workers aged <60 years, and the percentage of carbohydrates consumed was higher in both age groups. Agriculture is an energy-demanding occupation [45]. BMI was higher among agricultural workers under the age of 60, but body fat percentage was not significantly higher in agricultural workers. This suggests that agricultural workers did not have a high body fat percentage, which is an important indicator of health; hence, they might consume more calories because of their larger physiques. Therefore, excessive caloric restriction is not considered necessary in agricultural workers.

Ohta et al. reported the lifestyle of male agricultural, forestry, and other occupational workers in their 40s and 50s in Japan [11]. They reported that among agricultural and forestry workers, a little or hardly ever drinking habits and close friendships were more prevalent than among non-agricultural workers; moreover, the proportion of participants in this category practicing physical exercise was lower, while the number of household members and sleeping hours were higher than among other occupational workers. Our report shows a similar trend, although not as significant. Zhang et al. compared the health-promoting lifestyles of agricultural and non-agricultural workers in Japan [10]. In that study, spiritual growth was higher and physical activity was lower in the agricultural group than in the non-agricultural group, as evaluated using HPLP-II. In our study, the proportion of participants who had regular physical activity or exercise habits in their leisure time was low among agricultural workers. Similar reports of the low percentage of leisure time exercise among agricultural workers has been made in other countries, such as Australia and Poland [46,47]. Although our results are similar to those previously reported, Ohta et al. evaluated only men in their 40s and 50s, and Zheng et al. evaluated only the HPLP-II. This study expanded the scope to 1) include people aged ≥60 years and women; 2) evaluate many lifestyles; and 3) examine the details of bedtime and wake-up time using circular statistics [11].

In summary, to create effective interventions for people, the study results suggest that compared with non-agricultural workers, agricultural workers have different lifestyles in terms of sleep, work, nutrition, and time spent on each daily activity. The results of this study indicate that sleeping and napping, which are considered important for recovery from fatigue, were better among agricultural workers. Advising them to maintain satisfactory sleep habits may be helpful. However, it may be useful to consider further interventions, such as teaching programs that promote recovery in leisure time (e.g., to promote psychological detachment from work).

This study had some limitations. First, as a cross-sectional study, the causal relationship between agricultural workers and lifestyle cannot be assessed. Second, this study included participants from only one region; consequently, the results may differ from those of other regions. Third, participation was voluntary; the sample population may be more biased toward those who are interested in health promotion than the general population [23]. Fourth, in this

study, we were unable to assess the amount of time and workload performed when working as an agricultural worker. Finally, we did not examine the association between lifestyle differences and outcomes. Longitudinally designed investigations involving other regions are needed.

## Conclusions

Agricultural workers have better sleep habits but work longer hours compared with non-agricultural workers. Differences in energy intake and the proportion of time spent on each daily activity were noted. These differences should be considered when planning intervention programs for agricultural workers.

## Supporting information

**S1 Appendix. Table A. Proportion of participants who were engaged in each occupation.** (DOCX)

**S2 Appendix. Details of measurements.** (DOCX)

## Acknowledgments

We express our sincere gratitude to Professor Shigeyuki Nakaji for leading the Iwaki Promotion Health Project, and the staff of Hirosaki University for their contributions.

## Author Contributions

**Conceptualization:** Akihiro Hirakawa, Masahiro Nakatochi.

**Data curation:** Fumie Kinoshita, Kei Yoshida, Masaya Fujitani, Yuta Imai, Yumiko Kobayashi, Tomoya Ito, Yuto Okumura, Hiroyuki Sato, Akihiro Hirakawa.

**Formal analysis:** Fumie Kinoshita, Kei Yoshida, Masaya Fujitani, Yuta Imai, Yumiko Kobayashi, Tomoya Ito, Yuto Okumura, Hiroyuki Sato, Akihiro Hirakawa, Masahiro Nakatochi.

**Funding acquisition:** Fumie Kinoshita, Akihiro Hirakawa, Masahiro Nakatochi.

**Investigation:** Tatsuya Mikami, Songee Jung.

**Project administration:** Masahiro Nakatochi.

**Supervision:** Akihiro Hirakawa, Masahiro Nakatochi.

**Writing – original draft:** Fumie Kinoshita, Masahiro Nakatochi.

**Writing – review & editing:** Fumie Kinoshita, Kei Yoshida, Masaya Fujitani, Yuta Imai, Yumiko Kobayashi, Tomoya Ito, Yuto Okumura, Hiroyuki Sato, Tatsuya Mikami, Songee Jung, Akihiro Hirakawa, Masahiro Nakatochi.

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
