## [Decision Letter · Decision Letter 0]

11 Apr 2023

PONE-D-23-05965Lifestyle parameters of Japanese agricultural and non-agricultural workers aged 60 years or older and less than 60 years: A cross-sectional observational studyPLOS ONE

Dear Dr. Nakatochi,

Thank you for submitting your manuscript to PLOS ONE. After careful consideration, we feel that it has merit but does not fully meet PLOS ONE’s publication criteria as it currently stands. Therefore, we invite you to submit a revised version of the manuscript that addresses the points raised during the review process.

This paper needs significant revision. The introduction, methodology, and conclusions are disjointed and inconsistent. Please significantly increase the volume of each part to make it more readable to the reader with overall consistency. As you know, there is no word limit for this journal. As it is now, many readers will not be able to accept the claims of this paper. Especially for the discussion, please clarify why the discussion is the way it is, how it is connected to the introduction, and what you intend to imply. It is second-guessing the conclusion and there is little information to be gained. Clearly, the responsible party has failed to check this. Please review the paper properly again with the entire team.

We look forward to receiving your revised manuscript.

Kind regards,

Kenju Akai, Ph.D.Economics

Academic Editor

PLOS ONE

Journal Requirements:

Reviewers' comments:

Reviewer's Responses to Questions

**Comments to the Author**

1. Is the manuscript technically sound, and do the data support the conclusions?

Reviewer #1: No

Reviewer #2: Partly

2. Has the statistical analysis been performed appropriately and rigorously? 

Reviewer #1: No

Reviewer #2: No

3. Have the authors made all data underlying the findings in their manuscript fully available?

Reviewer #1: Yes

Reviewer #2: No

4. Is the manuscript presented in an intelligible fashion and written in standard English?

Reviewer #1: Yes

Reviewer #2: Yes

5. Review Comments to the Author

Reviewer #1: Discovering factors that maintain good health is an important issue in an aging Japan. Many people in Japan engage in agriculture both as a profession and as a hobby, and if health-promoting factors related to it are discovered, it would be a valuable finding, and I believe this perspective is very important.

However, I don’t understand the value of this study because the focus of what this study revealed was unclear

Major point

1. First of all

What is the novelty that this study reveals?

I understand that you have done an exploratory study of the differences in lifestyle of agricultural workers compared to non-agricultural workers. And the hypothesis is stated in Lines 72-73, but what did you identify in this study to prove it is “meaningful”?

It looks like there are no major problems with Method and Results in this study. However, if the main hypothesis is not clear, these results will not work well. I will mention some discussion on below.

2. Discussion Line 245-246

To argue that long-hour workers should improve the quality of their leisure time, it is necessary to clarify 1) what is the leisure time quality and 2) what would change for long-hour workers if they improve the quality of their leisure time. In this study, it is difficult to make these assertions because there is no data showing, for example, that workers who work longer hours are more fatigued.

3. Discussion Line 255-271

The finding that agricultural workers under the age of 60 have fewer sleep disturbances and nap more than non-agricultural workers seems most interesting in this study. It seems to be related to the characteristics of early to bed and early to rise. However, I recommend that you first perform a multivariate analysis to see if age, gender, or other factors may be acting as confounding factors in the degree of sleep disturbance.

4. Discussion Line 274-275

It is true that agricultural workers consume a lot of energy, but these workers under the age of 60 have a slightly higher BMI, so it is possible that they are not consuming enough energy.

5. Discussion Line 295-298

It is described as effective, but what kind of effect can we expect? And will the results of this study contribute in any way to that search?

【Minor point】

1. Discussion Line 249-251

Do agricultural worker have more pain?

Reviewer #2: Thank you for the opportunity to review your manuscript.

This study surveyed the lifestyle differences between agricultural workers and non-agricultural workers. It is important issue. I would appreciate some clarification on a small number of issues, as below:

Introduction

Line 67 You noted the agricultural workers have higher risk for mental health problems in line 67.

Mental health problems are associated with sleep disorders. How do you assess the mental health problems like depression, how they use medications for those problems?

Methods

Line 116-121 You noted logistic regression model and adjust for age, sex, and years of education.

I did not find the result about multivariable regression analyses.

Results Table 2

You noted agricultural workers have lower physical activity among leisure activities.

How do you assess the daily activity including agricultural work in a day or in a week? Agricultural work is included a part of some physical activities in some physical activity guidelines. Physical activity in a week is a critical information among those people’s health.

Discussion

Line 286-289

The participation rate in hobbies and amusements, and sports were significantly lower among agricultural workers.

You noted agricultural worker had lower regular physical activity or exercise habits.

Is this trend a Japanese feature or international trend?

If you do not assess the overall physical activity in a week, this description may be incorrect.

6. PLOS authors have the option to publish the peer review history of their article (what does this mean?). If published, this will include your full peer review and any attached files.

Reviewer #1: **Yes: **Takeshi Endo

Reviewer #2: No

---

## [Author Response · Author response to Decision Letter 0]

4 Jun 2023

Response to Editor’s comments

We would like to thank the editor and reviewers for the important points that were raised. We especially reviewed the Introduction and Discussion sections and have corrected the inconsistencies in the paper that you pointed out, while also responding to the reviewer's comments. The entire team, including two corresponding authors, also reviewed these revisions. We have also provided point-by-point responses to each reviewer’s comment below.

This paper needs significant revision. The introduction, methodology, and conclusions are disjointed and inconsistent. Please significantly increase the volume of each part to make it more readable to the reader with overall consistency. As you know, there is no word limit for this journal. As it is now, many readers will not be able to accept the claims of this paper. Especially for the discussion, please clarify why the discussion is the way it is, how it is connected to the introduction, and what you intend to imply. It is second-guessing the conclusion and there is little information to be gained. Clearly, the responsible party has failed to check this. Please review the paper properly again with the entire team.

Response to Journal Requirements

1.Please ensure that your manuscript meets PLOS ONE's style requirements, including those for file naming. The PLOS ONE style templates can be found at https://journals.plos.org/plosone/s/file?id=wjVg/PLOSOne_formatting_sample_main_body.pdf　and　https://journals.plos.org/plosone/s/file?id=ba62/PLOSOne_formatting_sample_title_authors_affaffiliati.pdf

We have re-checked the journal's submission rules and adapted the style. The paper has now been formatted as per the templates that you have provided.

There are restrictions for Data Availability due to ethical concerns. Thus, we do not plan to change our current description: “The datasets generated and/or analyzed during the current study are not publicly available due to ethical concerns, but will be made available from the Hirosaki University COI Program Institutional Data Access/Ethics Committee (contact via e-mail: coi@hirosaki-u.ac.jp) for researchers who meet the criteria for accessing confidential data. Researchers need to be approved by a research ethics review at the organization with which the researchers using the data are affiliated.”

Response to Reviewer #1’s comments

Discovering factors that maintain good health is an important issue in an aging Japan. Many people in Japan engage in agriculture both as a profession and as a hobby, and if health-promoting factors related to it are discovered, it would be a valuable finding, and I believe this perspective is very important.

However, I don’t understand the value of this study because the focus of what this study revealed was unclear

We deeply appreciate your valuable comments. I have responded to your points below.

Major point

1. First of all

What is the novelty that this study reveals?

I understand that you have done an exploratory study of the differences in lifestyle of agricultural workers compared to non-agricultural workers. And the hypothesis is stated in Lines 72-73, but what did you identify in this study to prove it is “meaningful”?

It looks like there are no major problems with Method and Results in this study. However, if the main hypothesis is not clear, these results will not work well. I will mention some discussion on below.

(Reply)

Similar studies examining agricultural workers’ lifestyles have been limited to men, those under 60 years of age, or their endpoints were limited (reference 10, 11 of the revised manuscript). The novelty of this study was that many lifestyle-related factors were compared, including older adults and women. This study produces results that is applicable to a wider population.

Since we position this study as a hypothesis-generating exploratory study, no hypothesis exists. It would be meaningful if we could find agricultural worker’s specific lifestyles from a wide range of lifestyle-related factors, as this would provide basic data for considering how the government and companies can intervene effectively. Such information will also be important when designing future prospective studies for assessing the effect of interventions. 

We have rewritten this point more clearly (in the Introduction lines 67-79, Method lines 86-89).

(Revised)

Page 4-5, lines 67-79 of the revised manuscript:

However, few reports have focused on multiple lifestyle factors in agricultural and non-agricultural workers in Japan. Ohta reported the differing lifestyles among occupational groups, including agricultural and forestry worker groups; however, only men in their 40s and 50s were evaluated [11]. Zheng reported a difference in health-promoting lifestyles but only evaluated these using the Health Promoting Lifestyle Profile II (HPLC-II) [10].

It would be meaningful to assess agricultural worker’s specific lifestyles across a wide range of working ages and genders, as this would provide basic data for considering how the government and companies can intervene effectively and designing future prospective studies to assess the effect of interventions. In this hypothesis-generating exploratory study, we investigated various lifestyle-related items in a rural Japanese population and compared them between agricultural and non-agricultural workers to guide future interventions for occupational workers. 

Page 5, lines 86-89 of the revised manuscript:

This hypothesis-generating cross-sectional study is part of an ongoing observational study called the “Iwaki Health Promotion Project” that began in 2005, which entails an annual large-scale health survey of the residents of Iwaki-ku (formerly Iwaki-cho) in Hirosaki City, Aomori Prefecture, Japan [23]. 

2. Discussion Line 245-246

To argue that long-hour workers should improve the quality of their leisure time, it is necessary to clarify 1) what is the leisure time quality and 2) what would change for long-hour workers if they improve the quality of their leisure time. In this study, it is difficult to make these assertions because there is no data showing, for example, that workers who work longer hours are more fatigued.

(Reply)

In our original manuscript, we described “leisure time quality” as a method of leisure that would reduce the fatigue caused by working effectively. Previous studies have stated that sleep is important for recovery from work-related fatigue, and a recovery training program for leisure time was being considered (e.g., Takahashi et al., J Occupational Safety and Health, 2014;7(1):23-30). Providing such programs and making effective use of leisure time is expected to lead to worker safety, recovery from fatigue, and health maintenance for agricultural workers. That is the point we originally wanted to discuss, so we have provided a description that deepens the discussion in lines 268-272.

As you pointed out, this study has not shown any relationship between long working hours and fatigue. However, as indicated in the Discussion lines 257-259, there are various previously studies indicating adverse effects of long working hours. In addition, if workers work for long hours, the time available for leisure is likely to be shorter, regardless of the degree of fatigue. Therefore, it is considered important to make effective use of that short leisure time. 

To clarify the meaning and purpose of leisure time quality, we have added these points to the manuscript (Discussion lines 259-262, 268-272).

(Revised)

Page 18, lines 259-262 of the revised manuscript:

However, Japanese agriculture often involves family businesses; thus, it is not easy to shorten working hours [16]. Hence, it is better to use one’s time outside of work efficiently, since it is limited. In particular, it is important to find ways to quickly and efficiently reduce fatigue. 

Page 18, lines 268-272 of the revised manuscript:

Previous reports have stated that sleep during rest periods between work is important for recovery from work-related fatigue, and a recovery training program was also being considered in Germany [40]. Providing such programs and making effective use of leisure time is expected to improve worker safety, recovery from fatigue, and health maintenance for agricultural workers.

3. Discussion Line 255-271

The finding that agricultural workers under the age of 60 have fewer sleep disturbances and nap more than non-agricultural workers seems most interesting in this study. It seems to be related to the characteristics of early to bed and early to rise. However, I recommend that you first perform a multivariate analysis to see if age, gender, or other factors may be acting as confounding factors in the degree of sleep disturbance.

(Reply)

We apologize for the lack of clarity, but the results presented are adjusted for age, gender, and educational background. We have clarified this in the Method lines 130-133. For greater clarity, we have also added that the “Beta value and p-value adjusted for age, sex, and years of education” to the footnotes of Tables 2 and 3, that the “Odds ratio and p-value adjusted for age, sex and years of education” to the footnote of Table 4, and that “The circular general linear model was performed by adjusting for age, sex, and years of education” to the footnote of Table 5.

4. Discussion Line 274-275

It is true that agricultural workers consume a lot of energy, but these workers under the age of 60 have a slightly higher BMI, so it is possible that they are not consuming enough energy.

(Reply)

As you pointed out, BMI was higher in agricultural workers under the age of 60. We performed additional analysis to compare the body fat percentage. This showed that body fat percentage was not significantly higher in agricultural workers, which suggests that they did not have a high body fat percentage. This is an important indicator of health, as they may consume more calories because of their larger physiques. Therefore, excessive caloric restriction is not considered necessary for agricultural workers. We added the method and result of body fat percentage into the Method lines 110-111, Table 1, and the Discussion lines 291-296. 

(Revised)

Page 6, lines 110-111 of the revised manuscript:

Body mass index (BMI) and body fat percentage were measured during the health checkup.

Page 19, lines 291-296 of the revised manuscript:

BMI was higher among agricultural workers under the age of 60, but body fat percentage was not significantly higher in agricultural workers. This suggests that agricultural workers did not have a high body fat percentage, which is an important indicator of health; hence, they might consume more calories because of their larger physiques. Therefore, excessive caloric restriction is not considered necessary in agricultural workers.

5. Discussion Line 295-298

It is described as effective, but what kind of effect can we expect? And will the results of this study contribute in any way to that search?

(Reply)

Yes, this was a vague description, so we have added a few specific recommendations based on your points in #2 into the Discussion lines 316-320. 

(Revised)

Page 20, lines 316-320 of the revised manuscript:

The results of this study indicate that sleeping and napping, which are considered important for recovery from fatigue, were better among agricultural workers. Advising them to maintain satisfactory sleep habits may be helpful. However, it may be useful to consider further interventions, such as teaching programs that promote recovery in leisure time (e.g., to promote psychological detachment from work). 

【Minor point】

1. Discussion Line 249-251

Do agricultural worker have more pain?

(Reply)

Since agriculture is physical labor, we considered that agricultural workers might have more physical pain. To demonstrate this, we performed an additional analysis to compare SF-36 in agricultural and non-agricultural workers because it evaluates bodily pain. We added the results into Tables 2 and 3, Method lines 112-116, Results lines 216-219, Discussion lines 263-264, and S2 Appendix. It was shown that the bodily pain score was significantly lower (i.e., more pain) in agricultural workers under the age of 60.

(Revised)

Pages 6-7, lines 112-116 of the revised manuscript:

Health-Related Quality of Life (HRQoL) was assessed using the Short Form Health Survey version 2 (SF-36 v2). The SF-36 is a widely used, self-administered health questionnaire, comprising of 36 items on eight subscales [24]. We used the Japanese version of SF-36. Each scale was represented by a score, with higher scores indicating higher HRQoL [25, 26]. 

Page 16, lines 216-219 of the revised manuscript:

The results of SF-36 showed that the bodily pain (BP) score was significantly lower for agricultural workers <60 years of age (beta=-2.1, 95% CI=-3.9–-0.2, p=0.029). The score for the role limitations attributable to emotional problems was significantly lower for agricultural workers ≥60 years of age (beta=-2.4, 95% CI=-4.6–-0.1, p=0.037).

Page 18, lines 263-264 of the revised manuscript:

An assessment of HRQoL data from the SF-36 showed that agricultural workers had greater BP than non-agricultural workers aged <60 years. 

S2 Appendix, Page 2

Measurement of health-related quality of life

The score for each subscale was calculated as 100 × (score − lowest score that the score can take) / (score range that the score can take) and was standardized by considering the Japanese population, with a mean of 50 and standard deviation of 10. Physical functioning; role limitations attributable to physical problems (RP), bodily pain (BP), and general health; and mental subscales, including vitality (VT), social functioning (SF), role limitations attributable to emotional problems (RE), and mental health were included in the questionnaire. The physical and mental component summary scores (PCS and MCS) were then calculated.

Response to Reviewer #2’s comments

Thank you for the opportunity to review your manuscript.

This study surveyed the lifestyle differences between agricultural workers and non-agricultural workers. It is important issue. I would appreciate some clarification on a small number of issues, as below:

We appreciate your constructive comments and recommendations. Please find our point-by-point response to each comment below.

Introduction

Line 67 You noted the agricultural workers have higher risk for mental health problems in line 67.

Mental health problems are associated with sleep disorders. How do you assess the mental health problems like depression, how they use medications for those problems?

(Reply)

To evaluate for mental health, we performed additional analysis to compare SF-36 in agricultural and non-agricultural workers. We added the results into Tables 2 and Table3, Method lines 112-116, and Result lines 216-219. Each scale of SF-36 was represented by a score, with higher scores indicating higher Health-Related Quality of Life. This showed that Role emotional was significantly lower in those over 60 years of age (beta=-2.4, 95% CI=-4.6–-0.1, p=0.037), who might have psychological problems in work and activities. However, the mental health and mental component summary were not particularly low in agricultural workers, which means that the Mental Health Quality of Life is not low in agricultural workers within this study population. 

Depression and medication details of mental health were not collected in this study. These details are not the purpose of this study, so we consider this an issue for future consideration.

(Revised)

Pages 6-7, lines 112-116 of the revised manuscript:

Health-Related Quality of Life (HRQoL) was assessed using the Short Form Health Survey version 2 (SF-36 v2). The SF-36 is a widely used, self-administered health questionnaire, comprising of 36 items on eight subscales [24]. We used the Japanese version of SF-36. Each scale was represented by a score, with higher scores indicating higher HRQoL [25, 26]. 

Page 16, lines 216-219 of the revised manuscript:

The results of SF-36 showed that the bodily pain (BP) score was significantly lower for agricultural workers <60 years of age (beta=-2.1, 95% CI=-3.9–-0.2, p=0.029). The score for the role limitations attributable to emotional problems was significantly lower for agricultural workers ≥60 years of age (beta=-2.4, 95% CI=-4.6–-0.1, p=0.037).

S2 Appendix, Page 2

Measurement of health-related quality of life

The score for each subscale was calculated as 100 × (score − lowest score that the score can take) / (score range that the score can take) and was standardized by considering the Japanese population, with a mean of 50 and standard deviation of 10. Physical functioning; role limitations attributable to physical problems (RP), bodily pain (BP), and general health; and mental subscales, including vitality (VT), social functioning (SF), role limitations attributable to emotional problems (RE), and mental health were included in the questionnaire. The physical and mental component summary scores (PCS and MCS) were then calculated.

Methods

Line 116-121 You noted logistic regression model and adjust for age, sex, and years of education.

I did not find the result about multivariable regression analyses.

(Reply)

We apologize for the lack of clarity, but the results presented are adjusted for age, gender, and educational background. We wrote about this in Method lines 130-133. For greater clarity, we have also added that the “Beta value and p-value adjusted for age, sex, and years of education” to the footnotes of Tables 2 and 3, that the “Odds ratio and p-value adjusted for age, sex, and years of education” to the footnote of Table 4, and that the “Circular general linear model was performed by adjusting for age, sex, and years of education” to the footnote of Table 5.

Results Table 2

You noted agricultural workers have lower physical activity among leisure activities.

How do you assess the daily activity including agricultural work in a day or in a week? Agricultural work is included a part of some physical activities in some physical activity guidelines. Physical activity in a week is a critical information among those people’s health.

(Reply)

We agree it is important to evaluate the amount of physical activity in work. In the Mets (which is frequently used to evaluate the activity load) the average exercise intensity of agricultural work is set at the same level as walking, gardening, and so forth, as daily life activities, while the maximum exercise intensity is set at the same level as aerobics and swimming as exercise (Med Sci Sports Exerc. 2011 Aug;43(8):1575-81. 2011 Aug;43(8):1575-81). However, in this study, we were unable to assess is the amount of time and workload performed while working as an agricultural worker. Further investigation is needed to determine the physical activity for this work.

However, some reports suggest that work-related and leisure-time physical activities should be considered separately to health (Holtermann et al., Eur Heart J,2021, 42(15): 1499-1511). In this study, we have at least demonstrated the fact that active leisure activity amounts are low in agricultural workers under the age of 60. 

This point is added in lines 325-327 of the manuscript as a limitation of this study.

(Revised)

Page 20, lines 325-327 of the revised manuscript:

Fourth, in this study, we were unable to assess the amount of time and workload performed when working as an agricultural worker. 

Discussion

Line 286-289

The participation rate in hobbies and amusements, and sports were significantly lower among agricultural workers.

You noted agricultural worker had lower regular physical activity or exercise habits.

Is this trend a Japanese feature or international trend?

If you do not assess the overall physical activity in a week, this description may be incorrect.

(Reply)

Similar studies regarding the low percentage of exercise during leisure time for agricultural workers has been made in other countries, such as Australia and Poland (https://farmerhealth.org.au/2021/04/28/why-exercise-is-important, Biernat et al, Int J Environ Res Public Health. 2019 Dec 27;17(1):208.). As you have pointed out, we evaluated the activity of leisure time only, so we have corrected the words “exercise habits” to “exercise habits in leisure time,” and similarity for the international trend in the Discussion lines 306-309.

(Revised)

Page 20, lines 306-309 of the revised manuscript:

In our study, the proportion of participants who had regular physical activity or exercise habits in their leisure time was low among agricultural workers. Similar reports of the low percentage of leisure time exercise among agricultural workers has been made in other countries, such as Australia and Poland [46, 47].

---

## [Decision Letter · Decision Letter 1]

24 Jul 2023

PONE-D-23-05965R1Lifestyle parameters of Japanese agricultural and non-agricultural workers aged 60 years or older and less than 60 years: A cross-sectional observational studyPLOS ONE

Dear Dr. Nakatochi,

Thank you for submitting your manuscript to PLOS ONE. After careful consideration, we feel that it has merit but does not fully meet PLOS ONE’s publication criteria as it currently stands. Therefore, we invite you to submit a revised version of the manuscript that addresses the points raised during the review process.

We look forward to receiving your revised manuscript.

Kind regards,

Kenju Akai, Ph.D.Economics

Academic Editor

PLOS ONE

Journal Requirements:

Reviewers' comments:

Reviewer's Responses to Questions

**Comments to the Author**

1. If the authors have adequately addressed your comments raised in a previous round of review and you feel that this manuscript is now acceptable for publication, you may indicate that here to bypass the “Comments to the Author” section, enter your conflict of interest statement in the “Confidential to Editor” section, and submit your "Accept" recommendation.

Reviewer #2: All comments have been addressed

Reviewer #3: All comments have been addressed

2. Is the manuscript technically sound, and do the data support the conclusions?

Reviewer #2: Yes

Reviewer #3: Yes

3. Has the statistical analysis been performed appropriately and rigorously? 

Reviewer #2: Yes

Reviewer #3: Yes

4. Have the authors made all data underlying the findings in their manuscript fully available?

Reviewer #2: No

Reviewer #3: Yes

5. Is the manuscript presented in an intelligible fashion and written in standard English?

Reviewer #2: Yes

Reviewer #3: Yes

6. Review Comments to the Author

Reviewer #2: Unclear points which I pointed out were mostly addressed. I have no further comment for this manuscript.

Reviewer #3: Dear Authors,

I have enjoyed reading your manuscript. here are few suggestions to make it better.

1) how did you assess "time spent on each daily activity". Please clarify this in the method section, then add a unit the proper unit of measurement within the tables where you report these results.

2) I suggest combining table 2 & 3 to be similar to table 4 for better prestation of the data

Thank you

7. PLOS authors have the option to publish the peer review history of their article (what does this mean?). If published, this will include your full peer review and any attached files.

Reviewer #2: No

Reviewer #3: No

---

## [Author Response · Author response to Decision Letter 1]

3 Aug 2023

Response to Editor’s comments

Thank you for submitting your manuscript to PLOS ONE. After careful consideration, we feel that it has merit but does not fully meet PLOS ONE’s publication criteria as it currently stands. Therefore, we invite you to submit a revised version of the manuscript that addresses the points raised during the review process.

Thank you for the comment and invitation to revise the manuscript once more. Our point-by-point responses to the reviewer comments for this round, for which we would like to show our deepest appreciation, are given below.

Response to Reviewer #2’s comments

Unclear points which I pointed out were mostly addressed. I have no further comment for this manuscript.

Thank you very much for your important comments and re-consideration of our paper for publication.

Response to Reviewer #3’s comments

I have enjoyed reading your manuscript. here are few suggestions to make it better.

We would like to humbly oblige your additional suggestions, as they were valuable and helped enhance the quality of our paper. We have responded to your points below.

Major point

1) how did you assess "time spent on each daily activity". Please clarify this in the method section, then add a unit the proper unit of measurement within the tables where you report these results.

(Reply)

We thank you very much for this comment. For the proportion of time spent on each daily activity, the number of days and hours per week that the participant spent performing each activity were collected through a questionnaire, and the percentage of time spent on each daily activity per week was calculated. We have added this explanation to the manuscript (Method, Measurement subsection, lines 113–116) and in the S2 Appendix, and have added the unit pertaining to this variable (%) to Table 2.

(Revisions)

Pages 6–7, lines 113–116 of the revised manuscript:

For the proportion of time spent on each daily activity, the number of days and hours per week that the participant spent performing each activity were collected through a questionnaire, and the percentage of time spent on each daily activity per week was calculated. Data were collected in 18 categories as described in the S2 Appendix.

Page 1 of the revised S2 Appendix:

Proportion of time spent in each daily activity encompassed 18 activity options: “Personal chores,” “Eating,” “Commuting to work or school,” “Travel,” “Schoolwork,” “Housework,” “Caring or nursing,” “Childcare,” “Shopping,” “Watching TV/listening to the radio/reading newspapers or magazines,” “Rest and relaxation,” “Learning/self-education/training,” “Hobbies and amusements,” “Sports,” “Volunteer activities,” “Socializing,” “Medical examination and treatment,” and “Other.” Time spent on “Schoolwork” was not analyzed because this study was designed for occupational workers.

2) I suggest combining table 2 & 3 to be similar to table 4 for better prestation of the data

(Reply)

Following your suggestion, Tables 2 and 3 were combined into a single Table 2 in the revised manuscript.

---

## [Decision Letter · Decision Letter 2]

13 Aug 2023

Lifestyle parameters of Japanese agricultural and non-agricultural workers aged 60 years or older and less than 60 years: A cross-sectional observational study

PONE-D-23-05965R2

Dear Dr. Nakatochi,

We’re pleased to inform you that your manuscript has been judged scientifically suitable for publication and will be formally accepted for publication once it meets all outstanding technical requirements.

Kind regards,

Kenju Akai, Ph.D.Economics

Academic Editor

PLOS ONE

Additional Editor Comments (optional):

Reviewers' comments:

Reviewer's Responses to Questions

**Comments to the Author**

1. If the authors have adequately addressed your comments raised in a previous round of review and you feel that this manuscript is now acceptable for publication, you may indicate that here to bypass the “Comments to the Author” section, enter your conflict of interest statement in the “Confidential to Editor” section, and submit your "Accept" recommendation.

Reviewer #3: All comments have been addressed

2. Is the manuscript technically sound, and do the data support the conclusions?

Reviewer #3: Yes

3. Has the statistical analysis been performed appropriately and rigorously? 

Reviewer #3: Yes

4. Have the authors made all data underlying the findings in their manuscript fully available?

Reviewer #3: Yes

5. Is the manuscript presented in an intelligible fashion and written in standard English?

Reviewer #3: Yes

6. Review Comments to the Author

Reviewer #3: all point I have mentioned have been addressed. I have no further comment for this manuscript. best of luck

7. PLOS authors have the option to publish the peer review history of their article (what does this mean?). If published, this will include your full peer review and any attached files.

Reviewer #3: No

---

## [Editor Report · Acceptance letter]

25 Sep 2023

PONE-D-23-05965R2 

Lifestyle parameters of Japanese agricultural and non-agricultural workers aged 60 years or older and less than 60 years: A cross-sectional observational study 

Dear Dr. Nakatochi:

I'm pleased to inform you that your manuscript has been deemed suitable for publication in PLOS ONE. Congratulations! Your manuscript is now with our production department. 

Kind regards, 

on behalf of

Dr. Kenju Akai 

Academic Editor

PLOS ONE